# Prevalence, Correlates, and Sequelae of Child Sexual Abuse (CSA) among Indigenous Canadians: Intersections of Ethnicity, Gender, and Socioeconomic Status

**DOI:** 10.3390/ijerph20095727

**Published:** 2023-05-05

**Authors:** L. Maaike Helmus, Ashley Kyne

**Affiliations:** Department of Criminology, Simon Fraser University, Burnaby, BC V5A 1S6, Canada

**Keywords:** sexual abuse, Indigenous, colonization, Canada, childhood maltreatment

## Abstract

Child sexual abuse (CSA) is a severe and concerning public-health problem globally, but some children are at higher risk of experiencing it. The harms caused by colonization and particularly the inter-generational legacy of residential schools would presumably increase the vulnerability of Indigenous children in former British colonies. Among 282 Indigenous participants in Canada recruited from Prime Panels, CSA was reported by 35% of boys, 50% of girls, and 57% of trans and gender non-conforming participants. These rates are substantially higher than global meta-analytic estimates (7.6% of boys and 18.0% of girls). There was evidence of intersectionality based on socioeconomic status. CSA was associated with a variety of other indicators of negative childhood experiences and significantly predicted numerous negative outcomes in adulthood, including mental-health issues (e.g., PTSD), unemployment, and criminal legal-system involvement. Sexual abuse of Indigenous Canadian children is a public-health crisis, and layers of marginalization (e.g., gender, social class) exacerbate this risk. Trauma-informed services to address the harms of colonization are severely needed, in line with recommendations from Canada’s Truth and Reconciliation Commission.

## 1. Introduction

Sexual abuse of children is a severe and concerning public-health problem globally. The behaviour can vary widely (e.g., relationship of abuser and victim, duration of abuse), as can its impact on the survivor (e.g., based on their age or other risk and resiliency factors). Negative consequences of child sexual abuse (CSA) can be far-reaching and diverse, including physical-health problems (e.g., chronic pain, high blood pressure, obesity), high-risk sexual behaviour (as well as low sexual satisfaction), mental-health difficulties (particularly self-harm), self-regulation problems (e.g., substance abuse), and lower income [1,2,3,4,5,6]. There is also evidence for the abused–abuser hypothesis, in that men who commit sexual offences against children have elevated rates of CSA themselves [7].

Not all children are at the same risk of being sexually abused. There is high co-occurrence between CSA and other forms of abuse and neglect [8,9]. Risk also differs significantly by gender. Stoltenborgh et al. conducted a comprehensive meta-analysis examining global prevalence rates of CSA (from 305 estimates) [10]. On average, 7.6% of boys worldwide experienced CSA, and 18.0% of girls. For girls, rates were lower in Asia and Europe. For boys, rates were higher in Africa. Research on trans and gender non-conforming children is only just starting to develop, but one study suggests rates of CSA are elevated among this heterogenous group [11]. Not surprisingly, rates of CSA can vary widely based on who you ask (e.g., sample characteristics) and how you ask it, but global estimates are nonetheless informative as a reference point. This study explored the prevalence, correlates, and sequelae of CSA among an under-researched population: Indigenous people residing in the land currently known as Canada. Research on Indigenous populations is particularly important given that the harms of colonization and its subsequent marginalization of Indigenous Peoples would plausibly increase the risk of CSA in these communities.

### 1.1. Background and Ongoing Consequences of Colonization

The term “Indigenous” denotes a diverse group of peoples who have been connected to the land since time immemorial [12]. On Turtle Island (refers to the continent of North America), this refers to inhabitants prior to European colonization. Of the approximately 1.67 million Indigenous Peoples in Canada, 58% identify as First Nations, 35.1% as Métis (In Canada, “Métis” refers to an ethnic identity that developed primarily from intermarriage between French and Indigenous peoples during the fur-trade era preceding British colonization), and 3.9% as Inuit [13]. However, there is considerable variability within these categories. For example, there are approximately 634 First Nations communities, embodying 50 distinct nations and over 50 Indigenous languages [14]. Although specific legislation, policies, and statistics differ across Canada, the United States, Australia, and New Zealand, there are more similarities than differences in the history and impact of colonization across these four former British colonies. Notably, the geographic dispersions among Indigenous nations, cultures, and languages on Turtle Island bears little resemblance to the modern border between Canada and the United States. Although a comprehensive description of the unique histories of Indigenous Peoples on Turtle Island is beyond the scope of this review, a brief summary will outline ongoing impacts of colonization, focusing on Canada but highlighting some parallels in the United States.

Although Indigenous Peoples constitute only 4.5% of the Canadian adult population, they are the fastest-growing population, increasing by 42.5% since 2006 [15]. According to the 2016 census, 52% of Indigenous Peoples live in urban areas and 40% of First Nations live on-reserve [13,14]. (Reserves are tracts of land that the British and Canadian government relegated Indigenous people to live in. In the United States, the comparable term is “reservation”.) Reserves are known for some of the lowest-quality living conditions in Canada, including ongoing issues of poverty, lack of social services, poor health outcomes, and inadequate/unsafe housing, including insufficient access to running water and electricity [16,17]. Native American reservations in the United States face similar problems [18].

It is important to remember that modern imposed Canadian government and culture is often at odds with the language, traditions, and culture of Indigenous Peoples. For example, colonization introduced a European culture that tended to be patriarchal, individual, and opportunistic, in contrast to many Indigenous communities that were historically much more matriarchal, egalitarian, and collectivist [16,19]. Additionally, not all Indigenous nations signed treaties or agreements, or ceded their territories, rights, or titles to the government.

One of the most harmful institutions of colonization is the residential-school system, which operated from 1831 to 1996 in Canada, and was explicitly modeled after a similar system (called Indian Boarding Schools) in the United States [16]. The primary objective of residential schools was to erase the children’s identity by “[killing] the Indian in the child” [20] (p. 47). The residential-school system was administered by the federal government and religious institutions. Methods of cultural genocide in the schools included shame, banning cultural expressions/symbols, physical/emotional/sexual abuse, neglect, and medical experimentation [21,22,23,24]. The impact of residential schools has gained increased attention after very recent discoveries in Canada of remains of over 1000 Indigenous children, with most deaths having gone undocumented [25]. This has prompted similar investigations and discoveries in the United States [26], indicating that previous data about residential schools underestimated the number of deaths, most of which were caused by combinations of abuse, neglect, and unsanitary conditions. This has contributed to the rise of a new social movement, Every Child Matters, to honour the victims of these schools in both Canada and the United States, to demand justice for Indigenous Peoples, and to increase awareness of this history and its reverberating impacts [27].

Although not all Indigenous Canadians in the 21st century have been in residential schools, many of their immediate and distant ancestors have, and the unaddressed harms and injustices of colonization and residential schools have contributed to intergenerational trauma among Indigenous Peoples as a broad community [16,24]. The Royal Commission on Aboriginal Peoples notes that the intergenerational trauma has contributed to elevated rates of family breakdown, negative child rearing, substance use, domestic violence, and suicide [28]. Because of colonization, Indigenous Peoples have disproportionately experienced low income, unemployment, lower educational attainment, substance abuse, loneliness, and community fragmentation [24,29].

This history has also contributed to over-representation in the criminal legal system and over-policing, which had led to mistrust of law enforcement among Indigenous communities [30]. In comparison to the general population, Indigenous Peoples, especially Indigenous women, are overrepresented as both perpetrators and victims of crime [31]; this has been referred to as “Canada’s national disgrace” [32]. Similar patterns can be observed in the United States, Australia, and New Zealand, where conviction or incarceration rates range from three to 10 times higher for Indigenous people compared non-Indigenous people [33,34,35]. This over-representation is complex and likely attributable to many factors, including biases in the system (e.g., over-policing), as well as higher rates of crime due to systemic disadvantage and disproportionate exposure to risk factors [36,37].

In grappling with the public-health impacts of colonization (and the largest class-action lawsuit in Canadian history), the Truth and Reconciliation Commission (TRC) of Canada was established in 2008 to document and recognize the direct and indirect impacts of the residential-school system. In its final report after a seven-year investigation, the TRC provided 94 Calls to Action for the federal, provincial, and territorial governments to improve conditions for Indigenous Peoples in Canada [24]. Action Items 18 and 19 call for all levels of Canadian government to acknowledge that Indigenous health in Canada is a direct result of Canadian government policies, and governments need to close health-outcome gaps between Indigenous and non-Indigenous communities, with a particular focus on factors such as suicide, mental health, addictions, child-health issues, and availability of appropriate health services. Call to Action Item 21 calls upon the federal government to provide funding for Indigenous healing centres to address the physical, mental, emotional, and spiritual harms caused by residential schools, particularly in Nunavut and the Northwest Territories.

### 1.2. Purpose of the Current Study

Much of the data relied upon by the Truth and Reconciliation Commission [24] is qualitative, which is necessary to honour the stories of Indigenous Peoples. However, we need more quantitative data on public-health issues among Indigenous Canadians. There are insufficient studies of child sexual abuse. This study explores rates of CSA among Indigenous Canadians, including an examination of intersectionality with gender and socioeconomic status, correlates of CSA (which may represent risk factors), and sequelae of CSA. This research can inform Action Items 18, 19, and 21 of Canada’s Reconciliation plan [24]. Given the harms of colonization and residential schools, we hypothesized that rates of CSA would be higher for Indigenous Canadians of all genders compared to global estimates. We expected rates to be higher for women than men, and highest for trans and gender non-conforming participants due to the intersectionality of marginalized identities. Consistent with previous research, we also expected CSA to be correlated with other indicators of adversity in childhood and with negative outcomes in adulthood (e.g., related to mental health and criminal-justice-system involvement).

This paper involved secondary analysis of a dataset collected for a project examining correlates of crime among Indigenous participants. Consequently, the choice of variables for this study was largely motivated by convenience based on what was available in the dataset, and the analyses should be considered relatively exploratory.

## 2. Materials and Methods

### 2.1. Sample

Participants were recruited online from Cloud Research using Prime Panels. Individuals registered in these panels agree to the conditions of compensation and can select which studies they would like to participate in. Prime Panels recruits from numerous sources where users can register to participate in studies in exchange for different types of compensation, including cash, charitable donations, or points (through some kind of financial service, redeemable for rewards). To be included, participants needed to have self-reported as Indigenous and reside in Canada. Data were collected in March 2022 and the survey was started by 434 participants. Twelve cases were deleted, as they did not identify as Indigenous. Given the sensitive nature of the questions and potential reluctance to answer honestly, we included a question at the end where participants could request that we not use their data (55 cases were deleted based on these requests). Four participants were removed because they withdrew consent after completing the survey. Five cases with duplicate IP addresses were deleted. Given that this study is part of a dataset used for examining correlates of crime [38], 75 cases were deleted for not indicating whether they had a charge or a conviction for a criminal offence. One case was deleted based on missing information regarding childhood sexual abuse. The final sample size for this study was 282.

Although many participants were deleted because they did not complete the questionnaire, inspection of the data suggested that this was likely often a case of starting the survey and then closing out of it early on (e.g., perhaps they were interrupted/distracted or grew bored). Those deleted because of insufficient information on key variables or duplicate IP addresses (*n* = 81) spent an average of 8.1 min on the survey (*SD* = 15.0) compared to 23.7 min for those who provided sufficient information (*SD* = 49.9, Cohen’s *d* = 0.35, 95% CI of 0.10 to 0.60). Roughly one third of those missing cases did not advance further than the first few demographic questions of the study, so there are limited data available to compare them to survey completers. Of those deleted cases that answered some demographic variables (*n* = 70), there were no significant differences from survey completers in terms of ethnicity (e.g., First Nations, Métis, etc.), employment status, or gender identification. Those who quit the survey early were, however, significantly younger (*M* = 28 years, *SD* = 11.9, *n* = 50) than those who completed the survey (*M* = 33.6 years, *SD* = 14.1, Cohen’s *d* = 0.39, 95% CI of 0.08 to 0.69).

Table 1 summarizes sample sizes and base rates or means of all variables examined in the study, as well as descriptive information. The average age of the participants was 33.6 years old (range of 14 to 77). More than half the participants identified as First Nations (56.0%). Most participants identified as women (57.9%) or men (32.5%), with the remaining identifying as Two-Spirit (3.9%), non-binary/gender-fluid/twin-spirit (5.0%), and other (0.7%). For those who identified as a man or woman (*n* = 253), 82.2% identified as cisgender and 2.1% identified as trans (15.7% identified as other; 11 left this blank). To classify gender with sufficient sample sizes for analyses, we combined these variables into cisgender men (*n* = 78; 27.9%), cisgender women (*n* = 153; 54.6%), and trans or gender non-conforming people (*n* = 49; 17.5%); two participants did not provide any information on gender to allow for classification. (There may be some error in these classifications. Individuals who identified as men or women but did not specify whether they considered themselves cisgender or trans were categorized as cisgender based on previous experience with crowdsourcing research.)

### 2.2. Survey and Procedure

The original purpose of this study was to develop a culturally informed questionnaire that would examine potential culturally specific or culturally salient correlates of criminal-legal-system involvement for Indigenous peoples [38]. The questionnaire examined childhood, family background, cultural identity, substance abuse, relationships, employment, experiences of racism/discrimination, and self-reported charges/convictions. The primary variable in this study is self-reported experiences of CSA. As this was not the original purpose of the survey, CSA was not defined for participants. The survey item was part of a section on childhood experiences and said, “I was sexually abused growing up.” Participants could respond “No” (53.2%), “Once” (14.9%), “Twice” (3.9%), or “Three or more times” (28.0%). Given the lack of specification of this variable (e.g., whether it is referring to instances or perpetrators), analyses were conducted on a dichotomized variable examining any experience of sexual abuse while growing up.

To keep the survey concise and simple, we did not attempt to account for temporal ordering of most variables. The first sections of the survey asked about childhood experiences (up until the age of 18), and later sections asked about adulthood (age 18 and older). Consequently, we can infer temporal relationships examining childhood and adulthood variables. However, within other childhood variables, we cannot discern whether CSA occurred during, before, or after other circumstances.

Participants who consented to participate completed the study online in Qualtrics, with the average participant completing the questionnaire in 23.7 min (*SD* = 49.9). The compensation options for participants consisted of a CAD 20 gift card to Amazon or an electronic transfer of CAD 20 to the participant’s bank account. Participants were additionally compensated through their online-survey-platform conditions (these compensations cannot be specified by the researchers).

### 2.3. Ethical Considerations for This Research

Firstly, we recognize that our own identities influence our experiences and how we conduct research and interpret data. This study was carried out by two female Canadian citizens, one of whom has South Pacific Indigenous ancestry, and the other has European ancestry. One of the authors has a Bachelor’s degree with a double major in Indigenous Studies and Criminology, whereas the other has a PhD in Forensic Psychology, trained with an exclusively quantitative approach to research.

Canadian ethical guidelines from the Tri-Council Policy Statement (TCPS2) requires additional considerations for research related to Indigenous Peoples. To allow for community engagement and collaborative discussions with Indigenous community members, we consulted with Indigenous individuals who, in some capacity, work with Indigenous Peoples in the criminal legal system or in a particular Canadian neighbourhood (well known for exceptionally high levels of drug use, homelessness, poverty, crime, mental illness, and sex work, and a high Indigenous population), or who teach in the Indigenous Studies department at the authors’ primary university. The purpose of the consultations was to seek input from diverse Indigenous perspectives about how to explore risk factors for crime among Indigenous Peoples. Consultants external to our university consisted of a range of people who have worked with justice-involved Indigenous Peoples (e.g., shelter workers, a judge, a former corrections employee), Elders, and formerly incarcerated individuals.

We emailed 16 prospective consultants (six internal to our university and 10 external) to provide feedback on our survey within 14 days (in any format that suited them). Nine individuals responded (three internal and six external) and expressed interest in consulting, but ultimately only eight submitted feedback. We provided consultants with an honorarium of CAD 50. Our survey was refined after the consultation process and subsequently received ethics approval from our university Research Ethics Board.

### 2.4. Overview of Analyses

We used the E/O index [39,40] to compare the CSA rates in the current sample to those that would be predicted from global meta-analytic estimates, stratified by gender (from [10]). The E/O index was selected because it provides both an effect-size measure as well as a null-hypothesis significance test, and is fairly intuitive as a ratio. Specifically, the E/O index is the ratio of the number of predicted survivors of CSA (E) divided by the observed number (O; method M_0_ from [41]). E/O values below 1 indicate that meta-analytic estimates predicted fewer CSA survivors than our data observed. Following Rockhill et al. [40], the 95% confidence intervals for the E/O index can be calculated using the Poisson variance for the logarithm of the observed number of cases (O):(1)$$95\%CI\left(\frac{E}{O}\right)=(E/O)exp(\pm 1.96\sqrt{1/O})$$

If the confidence interval for the E/O index does not include 1, that means the CSA rates in our sample are significantly different than the predicted values.

Most analyses examined the prediction of a dichotomous variable, so we used bivariate logistic regression [42]. The regression coefficients presented are odds ratios, which quantify how much the odds of the outcome increase with each one-point increase in the predictor. Odds ratios are statistically significant if the 95% confidence interval does not include 1. When the outcome was assessed using variables rated on a Likert scale, Kendall’s tau correlations for ordinal data were reported. Given the observed intersectionality with our childhood-socioeconomic-status variable, in addition to bivariate-odds ratios, we also present ratios controlling for socioeconomic status.

## 3. Results

### 3.1. Prevalence of CSA and Intersectionality of Gender Identity and Socioeconomic Status

In the overall sample (*n* = 282), 132 participants (46.8%, 95% CI [42.8, 54.4]) self-reported experiencing CSA. The global meta-analysis [10] would have estimated 7.6% among boys and 18.0% among girls. We could not find a global meta-analytic estimate of CSA for trans and gender non-conforming children, but previous research suggests elevated rates [11], so they would be expected to be minimally as high as the estimates for girls (this was used as an admittedly lower-bound estimate). The rates in our sample were 34.6% among cisgender men (95% CI [24.0, 45.2]), 50.3% for cisgender women (95% CI [42.4, 58.2]), and 57.1% among trans and gender non-conforming participants (95% CI [43.2, 71.0]). Table 2 presents our observed number of CSA survivors by gender (3 categories), as well as the number that would be expected based on the meta-analytic estimates. All three E/O indices were statistically significant (0.22 for men, 0.36 for women, and 0.18 for trans and gender non-conforming), indicating that the global estimates of CSA were roughly one fifth to one third the rate that was observed in this Indigenous Canadian sample. Reversing the indices to provide a ratio of observed to expected CSA survivors, the rates of CSA were roughly 2.8 to 5.6 times higher than expected.

Living on a reserve was not associated with a higher rate of CSA (OR = 1.232, 95% CI of 0.740 to 2.050). To further explore intersectionality based on socioeconomic status, we summed three childhood items rated on a three-point scale (no/sometimes/yes): “I feel that my basic needs (e.g., food, clothing, housing, access to clean drinking water) were met growing up” (reverse scored), “I missed out on participating in activities or experiences because I did not have enough money,” and “I or my family relied on any charities, churches, or government services (e.g., welfare) for additional support.” Total scores on this summed variable ranged from 3 to 8. Each one-point increase for this item (reflecting lower socioeconomic status) was associated with a 71% increase in the odds of CSA (OR = 1.709, 95% CI of 1.423 to 2.052).

### 3.2. Correlates of CSA

This section examined other characteristics of participants’ childhood and family experiences that may predict CSA (but we cannot confirm temporal ordering for most variables). Table 3 presents the prevalence rates for these correlates (all dichotomously scored) and the odds ratios for their bivariate association with CSA, as well as odds ratios controlling for socioeconomic status. Odds ratios in text are for the bivariate associations and typically remained significant after controlling for socioeconomic status (unless specified otherwise). Significantly higher rates of CSA were reported among participants who had also been emotionally abused, who had been physically abused, whose primary caregiver had had a substance-abuse problem, and who had been placed in foster care (OR between 2.4 and 12.4). The largest relationships by far were with emotional abuse (OR = 12.4) and physical abuse (OR = 9.8). Notably, the base rates of these characteristics were also very high; over half of participants had been physically abused and two thirds had been emotionally abused. A total of 51% reported that their primary caregiver had had a substance-abuse problem, and 26% had been placed in a group home or foster care. Conversely, rates of CSA were significantly lower among participants who had felt loved by at least one caregiver (OR = 0.38, base rate = 91%; this effect was no longer significant after controlling for socioeconomic status: OR = 0.58) and who had felt supported in their well-being and mental health (OR = 0.15; base rate = 45%). Being adopted was not significantly related to CSA (OR = 0.62), although only 8% of participants were adopted, which would reduce statistical power.

Having attended residential school was not significantly associated with CSA (OR = 1.9), although only 10% of participants had been sent to residential school, reducing statistical power. In contrast, one third of participants reported that at least one of their biological parents had been sent to a residential school, and two thirds reported that at least one family member had been sent. Having a biological parent or other family member who had been sent to a residential school was significantly associated with higher rates of CSA among participants (ORs of 1.8 and 2.6, respectively), although the association for biological parents was no longer significant after controlling for socioeconomic status (OR decreased from 1.8 to 1.5). Notably, the effect size was similar for having personally attended residential school versus having a biological parent who had attended.

### 3.3. Sequelae of CSA

Table 4 examines outcomes in adulthood (or lifetime) that could presumably be consequences of CSA. Regarding criminal behaviour, roughly one third of participants self-reported having been charged with or convicted of any type of criminal offence (base rates are less than 20% for specific offence types examined, including drug possession, violent offences, and sexual offences). CSA survivors were more likely to be convicted of any type of offence (OR = 1.9), but CSA did not specifically predict drug possession, violent, or sexual offences (ORs between 0.91 and 1.7). Additionally, the relationship with any charge/conviction was no longer significant after controlling for socioeconomic status (OR = 1.6).

Just over half of the participants self-reported having attempted suicide at any point in their life, with higher rates among CSA survivors (OR = 4.4). We combined questions about suicide attempts in childhood and adulthood because we presumed that if the suicide attempt was related to the CSA, there was a good chance it had occurred before the age of 18. However, it is possible in some cases that the suicide attempts preceded CSA. When restricting the variable only to suicide attempts in adulthood (39% base rate), rates were still significantly higher for CSA survivors, with a slightly reduced odds ratio of 3.5. CSA significantly predicted other variables related to mental health, including diagnosis of PTSD (OR = 4.2; base rate = 36%), regular use of alcohol at some point in life (OR = 3.9; base rate = 69%), use of illegal drugs (OR = 2.6; base rate = 61%), and current self-reported substance-abuse problems (OR = 2.4; base rate = 29%). CSA also significantly predicted having dropped out of high school (OR = 2.2; base rate = 37%) and lower rates of employment (OR = 0.61; base rate = 46%), but these latter two effects were no longer significant after controlling for socioeconomic status in childhood.

Three non-dichotomous variables were examined to assess positive functioning among participants: feelings of life satisfaction, feelings of control over life, and feeling capable of tackling problems/stressors. CSA was negatively correlated with life satisfaction (tau = −0.184, *p* < 0.001), feelings of control (tau = −0.143, *p* = 0.010), and feeling capable of tackling problems (tau = −0.113, *p* = 0.043). Effect sizes were small in magnitude. Using partial correlations (Pearson’s) to control for socioeconomic status, the corrections reduced to non-significance (life satisfaction *r* = −0.117, *p* = 0.0502; control *r* = −0.074, *p* = 0.220; capable *r* = −0.041, *p* = 0.490).

## 4. Discussion

This study found that, depending on gender, rates of CSA experienced by Indigenous Canadians were roughly three to five times higher than global estimates. There was evidence of intersectionality based on socioeconomic status, although there were no differences between Indigenous Canadians who lived on- or off-reserve during their childhood. CSA was associated with a variety of other indicators of negative childhood experiences and significantly predicted numerous negative outcomes in adulthood, including mental-health issues (e.g., PTSD), unemployment, and criminal-legal-system involvement.

Indigenous children are exceptionally vulnerable to CSA; roughly half of participants experienced CSA. This is a public-health crisis in need of immediate, targeted attention. Drawing comparisons to global meta-analyses is challenging given methodological differences, especially as prevalence rates are likely to be impacted by how the questions are defined and asked. Prevalence rates should be higher when participants are given behavioural indicators (e.g., unwanted touching). In this study, we simply asked whether they had been sexually abused; we did not define sexual abuse. This requires that the participant identify and label their experience as sexual abuse. A meta-analysis found that over half of women who experience rape at the age of 14 + do not label their experience as rape [43]; instead, they define the event with more benign and self-blaming narratives. Consequently, it is unlikely that the methodology of our study would inflate the prevalence of CSA compared to global estimates. If anything, the phrasing of our question would underestimate sexual-abuse rates.

Prevalence rates of CSA also depend on who you ask. Clinical samples would likely have higher rates of CSA. University samples would likely have lower rates, as it would screen for more privileged and higher-functioning individuals, on average. Studies that advertise as being about CSA may garner more interest and responses from people who have experienced CSA. The current study used online research panels from a larger study that was advertised as being about understanding why some Indigenous people commit crimes and some do not. The way the participants were recruited should not have substantially preselected individuals more likely to have been sexually abused. It is possible that the study was more likely to appeal to Indigenous people who have a criminal record, which could have inflated the proportion of justice-involved individuals in our study. We found that those with a criminal record were more likely to have been sexually abused, so there could be a small participant-selection bias at play that may have indirectly inflated the CSA estimates. However, we do not expect this would have had a large impact on the results, and certainly not enough to explain the drastically higher rates of CSA in the current sample compared to global estimates.

It is important to consider the likelihood that our study is representative of Indigenous Canadians. Our sample would presumably be at least somewhat skewed towards people more educated (e.g., literate), English-speaking (e.g., excluding Francophones), younger (e.g., internet access), and unemployed (e.g., have time to complete surveys). These limitations are similar to virtually all research conducted on humans, with the exception of basic census data that are mandated by law. Even most putatively nationally representative surveys are often not fully representative based on requiring a phone or a permanent address, linguistic fluency, intellectual competence, and time/motivation to participate.

We have no reason to expect systematic biases in our sample compared to Indigenous Canadians that would meaningfully impact our results. Participants from online panels generally tend to answer reliably and consistently [44]. However, online pools are not perfectly representative of the general population. For example, MTurk users are more educated, less religious, and more likely to be unemployed than the general population [45]. In contrast, participants recruited from Prime Panels (the source used for the current study) tend to be more diverse than participants in MTurk [46].

Common with online survey panels, our study disproportionately sampled women compared to men. Official data in Canada suggest that roughly 51.9% of Indigenous Canadians are female, 47.4% are male, and 0.6% are trans or non-binary [47]. Our data comprised 55% cis women, 28% cis men, and 18% trans or gender non-conforming individuals. Our greatly elevated rates of gender non-conforming individuals may reflect our inclusion of the option to identify as two-spirit. Although our gender composition is clearly not representative of the Canadian population, our comparisons to global prevalence rates of CSA were gender stratified. We have no reason to expect the relative relationships examined in this paper (correlates and consequences of CSA) would be impacted strongly by gender, although this should be examined in future research with larger sample sizes. Our sample does appear to be roughly representative of the education levels of Indigenous Canadians. A total of 37% of participants reported having dropped out of high school, which is identical to 2017 estimates for Indigenous Canadians, although these statistics examined only Indigenous individuals living off-reserve [48].

To the extent that this survey may have been disproportionately completed by individuals with a criminal history and who may be unemployed, this could slightly inflate the absolute rates of CSA reported. However, relative relationships between variables are less likely to be impacted by these differences. Additionally, our unusually high rates of compensation to participants for this research study hopefully incentivized a larger and more diverse participant pool. Lastly, the most likely differences in our sample (e.g., having a criminal record and being unemployed) tended to have smaller relationships with CSA.

Based on our earlier review of the context of Indigenous Peoples in Canada, we believe that the leading explanation for the high rates of CSA in Indigenous communities is inextricably linked to the harms of colonization. Disentangling the precise causes of this public-health problem is difficult, as the legacy of colonization and racism has contributed to a host of adverse circumstances disproportionately facing Indigenous populations, many of which would overlap with risk factors for numerous negative outcomes, including sexual abuse. Consequently, there are likely several mediators at play. Nonetheless, we believe colonization is a key distal contributor to these findings but not necessarily the only contributor. Alternative explanations would likely be rooted in some assertion of cultural differences whereby Indigenous people are more likely to commit CSA. We are not aware of any evidence or theory to support such explanations; if anything, data on people who have committed sexual offences suggest lower levels of sexual deviance (including pedophilic behaviours) among Indigenous perpetrators compared to non-Indigenous perpetrators [49,50].

Contrary to expectations from previous research [7], we did not find evidence of a direct abused–abuser link in terms of CSA survivors being more likely to be charged or convicted of a sexual offence. However, it is possible that this pattern is uniquely male, as men represent roughly 90% of sexual-abuse perpetrators [51]. Only a third of study participants identified as male; future research examining larger samples of male offenders should examine this relationship in more detail. Additionally, our small sample size of people charged with a sexual offence also reduced statistical power for this analysis.

Our study found that Indigenous Canadians whose parents or extended family members had attended residential schools were significantly more likely to experience CSA. These effects were similar to or larger than the effect size for personally having attended a residential school. This finding confirms the intergenerational trauma caused by residential schools [16,24,28]. The precise mechanism through which familial experience of residential schools increases risk of CSA is not known, but there are numerous possibilities that could contribute, likely in tandem. The most direct explanation is an abused–abuser hypothesis [7], whereby Indigenous peoples who had attended the schools and been abused themselves as children are more likely to go on to abuse future generations of Indigenous children, particularly those in their own family. In addition to abuse experienced by residential-school survivors, the schools have also contributed to elevated rates of community fragmentation, substance abuse, and mental-health problems [16,28], all of which could serve as fertile ground for increasing the risk of survivors perpetrating diverse types of violence, regardless of whether that individual was abused in the schools. Additionally, children, nieces/nephews, and grandchildren of residential-school survivors may be at elevated risk not just from their own family members who had attended residential schools, but from other community members (Indigenous or non-Indigenous). Attendance in residential schools can cause diverse trauma responses among survivors, which could serve to weaken parenting skills or the ability to effectively monitor their children’s whereabouts, increasing the vulnerability of children to other perpetrators in the community. For example, Indigenous children are more likely to be placed in foster care [16]; it is likely that descendants of residential-school survivors may be disproportionately placed in foster care, elevating their risk of abuse by foster parents.

Although the evidence of intergenerational trauma is clear, the dynamics of it are not fully understood, nor are the optimal solutions. Nonetheless, trauma-informed approaches may be particularly useful in providing services to Indigenous Peoples [52]. Practitioners should be particularly cognizant of ways in which government systems, even those that are designed to offer assistance, may bear similarities with systems that have oppressed and devalued Indigenous peoples and instilled understandable distrust among Indigenous communities.

Future research should examine some of these findings in more detail, and larger samples would be necessary to separate analyses by gender. Larger studies could better examine patterns of CSA over time and identify additional risk and resiliency factors. This study used a dataset developed for a different research question. A study focused on CSA could utilize better definitions with specific behavioural definitions and measure more specific features of the CSA, including number of perpetrators and the age(s) during which abuse occurred.

### Recommendations for Policy and Practice

There is a variety of factors that likely contribute to the vulnerability of Indigenous communities to adverse conditions, including the devastating effects of colonization and forced assimilation, social and economic marginalization, the extraction of resources, cultural and spiritual isolation, and discriminatory policies and practices upheld by the government and law enforcement [16,21,23,53,54]. Unfortunately, governments of the former British colonies have typically taken insufficient accountability for the problems their policies have caused. Contrary to what is often claimed, colonization is still alive and well, and policy changes have rarely delivered on their promises to promote the best interests of Indigenous peoples.

The way forward in addressing these issues is to advance reconciliation efforts, consistent with recommendations such as those already specified in the Truth and Reconciliation Commission of Canada [24], such as providing funding for Indigenous healing centres. Given the astronomically high rates of CSA in Indigenous communities reported in the current study, as well as high rates of caregiver substance-abuse problems, suicidal behaviour, and placements in group homes or foster care, health providers in Canada should be better educated on these issues and may benefit from taking a trauma-informed approach in providing services to Indigenous individuals [52].

Given the numerous sovereign Nations, tribes, and communities in Canada and other former British colonies, there cannot be a one-size-fits-all plan for change. Nonetheless, engaging in cultural activities and building a cultural identity may be critical to the healing process, as it can promote adaptive coping and well-being [55]. Cultivating a new relationship with Indigenous Peoples must involve respecting their right to self-determination, including developing community-driven resources for healing. We (i.e., descendants of settlers) must also decolonize our thinking patterns by recognizing the role of colonialism in our lives and how it shapes our communities. Decolonization, however, involves Indigenous Peoples reclaiming rights and sovereignty to promote humanity, freedom, and independence for Indigenous peoples [56].

## 5. Conclusions

Sexual abuse of Indigenous Canadian children is a public-health crisis, with rates approximately three to five times higher than global estimates. The harms of colonization have placed Indigenous children at particular risk of abuse, and layers of marginalization (e.g., gender, social class) exacerbate this risk. Trans and gender non-conforming Indigenous participants had exceptionally high rates of CSA (57%). The study sample was Canadian, but given similarities in the consequences of colonization in Canada, the United States, Australia, and New Zealand, we would expect similar patterns in those countries, as well. We are often told that research on Indigenous Canadians would not generalize to other countries because of the variability in Indigenous cultures and languages. However, there is no reason to believe that these findings are related to Canadian Indigenous cultural factors. They are most plausibly attributed to the legacy of colonization. Although research is needed to confirm this, we suspect that the public-health and criminal-justice problems caused by colonization are remarkably similar across these four former British colonies, despite variability in Indigenous cultures. That being said, the optimal solutions and paths to healing should take into account the variability in cultures between and within these countries. In other words, the harms of colonization are fairly generalizable; the solutions, however, need to be grounded in the particular cultural context of the nations involved in order to chart a path forward. This study supports Canada’s Truth and Reconciliation Commission’s Calls to Action to close health-outcome gaps between Indigenous and non-Indigenous communities and provide funding for Indigenous healing centres to address the harms caused by residential schools (Action Items 18, 19, and 21; TRC, 2015).

## Figures and Tables

**Table 1 ijerph-20-05727-t001:** Descriptive information for all variables analyzed.

Variable	N with Data	*n* (%) with Characteristic (Dichotomous Variables)	M	SD
Gender	280			
Cisgender men		78 (27.9)		
Cisgender women		153 (54.6)		
Trans or gender non-conforming		49 (17.5)		
Age	276		33.6	14.1
Ethnicity	282			
First Nations		158 (56.0)		
Metis		98 (31.6)		
Inuit		5 (1.8)		
Native American		14 (5.0)		
Mixed		13 (4.6)		
Other		3 (1.1)		
Experienced CSA	282	132 (46.8)		
Lived on a reserve growing up	271	88 (32.5)		
Lower SES (sum of 3 variables, possible range of 3–9) ^a^	281		5.5	1.5
Emotionally abused	282	187 (66.3)		
Physically abused	282	158 (56.0)		
Felt loved by at least one caregiver	282	257 (91.1)		
Primary caregiver had substance-abuse problem	282	143 (50.7)		
Felt supported in well-being and mental health	282	128 (45.4)		
Went to residential school	282	28 (9.9)		
Any family member went to residential school	282	195 (69.1)		
Biological parent went to residential school	260	82 (31.5)		
Placed in group home or foster care	268	70 (26.1)		
Adopted	277	22 (7.9)		
Charged with or convicted of any offence	282	88 (31.2		
Charged with or convicted of possession of drugs	282	34 (12.1)		
Charged with or convicted of violent offence	282	47 (16.7)		
Charged with or convicted of sexual offence	282	9 (3.2)		
Suicide attempt (lifetime)	282	148 (52.5)		
Suicide attempt (adulthood)	282	110 (39.0)		
Diagnosed with PTSD	245	89 (36.3)		
Used alcohol regularly at some point in life	281	194 (69.0)		
Used illegal drugs	277	169 (61.0)		
Current substance-abuse problem	282	82 (29.1)		
Dropped out of high school	279	104 (37.3)		
Employed	281	130 (46.3)		
I currently feel satisfied with my life (1–4 scale)	282		3.0	0.9
I feel like I have control over my life (1–4 scale)	281		2.9	0.9
I feel capable of tackling problems or stressors in my life (1–4 scale)	282		3.0	0.9

^a^ Sum of 3 childhood items rated on a 3-point scale (no/sometimes/yes): “I feel that my basic needs (e.g., food, clothing, housing, access to clean drinking water) were met growing up” (reverse scored), “I missed out on participating in activities or experiences because I did not have enough money”, and “I or my family relied on any charities, churches, or government services (e.g., welfare) for additional support”.

**Table 2 ijerph-20-05727-t002:** E/O comparing expected and observed number of CSA survivors.

		Expected CSA Rates		Observed	Effect Size					
Gender of Participant	N	%	N	CSA N	E/O	LL	UL	O/E	LL	UL
Cisgender men	78	7.6%	5.9	27	0.22	0.15	0.32	4.6	3.1	6.7
Cisgender woman	153	18.0%	27.5	77	0.36	0.29	0.45	2.8	2.2	3.4
Trans or gender non-conforming	49	18.0%	5.0	28	0.18	0.12	0.26	5.6	3.8	5.6

Note: E/O = expected/observed. LL = lower limit of 95% confidence interval. UL = upper limit.

**Table 3 ijerph-20-05727-t003:** Correlates of CSA (childhood and family-background variables).

	Unadjusted Odds Ratios			Adjusted for Low SES			
Predictor Variable	OR	LL	UL	SES Sig.	OR	LL	UL
Emotionally abused	**12.388**	**6.307**	**24.333**	**Y**	**9.082**	**4.534**	**18.190**
Physically abused	**9.768**	**5.555**	**17.179**	**Y**	**7.266**	**4.043**	**13.057**
Felt loved by at least one caregiver	**0.381**	**0.159**	**0.915**	**Y**	0.579	0.233	1.439
Primary caregiver had substance-abuse problem	**2.869**	**1.767**	**4.658**	**Y**	**1.871**	**1.104**	**3.170**
Felt supported in well-being and mental health	**0.145**	**0.085**	**0.247**	**Y**	**0.206**	**0.117**	**0.361**
Went to residential school	1.868	0.841	4.147	Y	1.757	0.744	4.153
Any family member went to residential school	**2.593**	**1.519**	**4.424**	**Y**	**1.902**	**1.072**	**3.376**
Biological parent went to residential school	**1.844**	**1.086**	**3.131**	**Y**	1.485	0.838	2.630
Placed in group home or foster care	**2.399**	**1.372**	**4.196**	**Y**	**1.923**	**1.058**	**3.498**
Adopted	0.623	0.253	1.536	Y	0.622	0.236	1.641

Note: All variables are dichotomous. OR = odds ratio. LL = lower limit of 95% confidence interval. UL = upper limit. Sig. = significant. Y = yes. N = no. Bolded values are statistically significant (*p* < 0.05).

**Table 4 ijerph-20-05727-t004:** Sequelae of CSA (adulthood and lifetime variables).

	Unadjusted Odds Ratios			Adjusted for Low SES			
Outcome Variable	OR	SES Sig.	SES Sig.	SES Sig.	OR	LL	UL
Charged with or convicted of any offence	**1.923**	**1.154**	**3.204**	**N**	1.591	0.921	2.747
Charged with or convicted of possession of drugs	1.513	0.735	3.114	Y	1.082	0.500	2.342
Charged with or convicted of violent offence	1.671	0.888	3.147	N	1.381	0.699	2.726
Charged with or convicted of sexual offence	0.906	0.238	3.448	N	0.612	0.149	2.523
Suicide attempt (lifetime)	**4.398**	**2.659**	**7.273**	**Y**	**3.608**	**2.122**	**6.134**
Suicide attempt (adulthood)	**3.537**	**2.139**	**5.847**	**Y**	**3.689**	**2.164**	**6.290**
Diagnosed with PTSD	**4.210**	**2.422**	**7.318**	**N**	**3.641**	**2.041**	**6.493**
Used alcohol regularly at some point in life	**3.869**	**2.209**	**6.778**	**N**	**3.873**	**2.134**	**7.029**
Used illegal drugs	**2.552**	**1.542**	**4.226**	**Y**	**1.993**	**1.165**	**3.410**
Current substance-abuse problem	**2.417**	**1.425**	**4.099**	**N**	**2.228**	**1.269**	**3.913**
Dropped out of high school	**2.214**	**1.350**	**3.632**	**Y**	1.540	0.901	2.632
Employed	**0.608**	**0.378**	**0.976**	**N**	0.680	0.409	1.131

Note: All variables are dichotomous, with CSA as the predictor variable. OR = odds ratio. LL = lower limit of 95% confidence interval. UL = upper limit. PTSD = post-traumatic stress disorder. Sig. = significant. Y = yes. N = no. Bolded values are statistically significant (*p* < 0.05).

## Data Availability

Analyses can be verified at Simon Fraser University by contacting maaike_helmus@sfu.ca.

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
