# Peer review of "Prevalence, Correlates, and Sequelae of Child Sexual Abuse (CSA) among Indigenous Canadians: Intersections of Ethnicity, Gender, and Socioeconomic Status"

_ijerph, 2023, doi:10.3390/ijerph20095727_

Round 1

Reviewer 1 Report

Thank you very much for providing me with an opportunity to review this manuscript.

The manuscript topic is important, because the population under investigation is under researched and neglected and the authors have presented the content in a clear way. Literature indicates that most of the studies are qualitative on the present sample however; the present study is quantitative in nature and it is highlighting the severity of the issue compared to global estimates. The findings of the present study indicate that CSA is significantly higher in the current population than the global estimates and therefore, it is important for researchers to continue to explore the salient factors related to prevalence, correlates, and sequelae of CSA in the current population to further confirm the present findings. The higher prevalence of CSA in the present population also calls attention from researchers and policy makes to explore and highlight biases in the system and establish systems that will decrease their psychological issues, prevent further deterioration of the mental well being and improve conditions for Indigenous populations.

The present research study topic, population, method of analysis and writing style area appropriate. This research is a steppingstone for future research studies on the same topic because the results of the present study suggests a higher prevalence of CSA in indigenous population compared to global estimate.

I would highly recommend publishing this  manuscript as it is because I did not find any major fault that needs to be addressed before publishing. 

Author Response

Thank you for your kind comments.

Reviewer 2 Report

Overall assessment:

This paper handles the important and understudied subject of CSA among indigenous people in Canada. The research questions are well motivated, and the authors use unique data. However, the framing of the study could be more clear and the method could be better motivated. I also miss some more reasoning in the conclusions about how these are reached.

Framing/text:

When reading the introduction, I was led to believe that the article would be about the consequences of attending residential schools. In the results section it is however revealed that only 10% of the sample attended. You should reframe your introduction and background to better reflect that you investigate childhood CSA for indigenous persons in general, not just those that went to the schools. 

Even without any experience as a first nation or of CSA, I found the text heavy to read and at times triggering. Your paper may be read by persons who have personally been affected by the subject at hand. Is there a way you can include links/phone numbers to resources for those struggling with these types of experiences?

You mention that your survey was framed as an effort to understand why indigenous persons are more often charged with crimes and that this may lead to selection bias in your sample. What is the proportion of indigenous persons who are ever charged or convicted of any offence in Canada? Is this proportion the same as in your sample? If so that speaks against any selection bias.

You find a correlation between family members attending residential schools and CSA. Why do you think this is? I would have liked a more elaborate discussion on the psychological and theoretical explanations for why this may be.

You seem to jump to the conclusion that colonialization is the culprit behind the problems in the indigenous community. This may be, but I think you could develop further how you reach this conclusion, based on your results.

Data and method:

From your data description it was not clear how you classify the 11 persons who did not answer if they were cis gender or not. Should footnote 4 be understood as that they were classified as cisgender?

The number of observations is stated as "cisgender men (n = 78; 27.9%), cis-gender women (n = 153; 54.6%), and trans or gender non-conforming people (n = 49)" This sums to 280. How were the remaining 2 persons classified?

The data section lacks information on what other variables are included in the data other than age, gender and nationality.

The authors explain what the e/o index is, but do not motivate why this is the best suited way of evaluating risk of CSA in their sample.

The article is very short. The authors could have spent more time motivating their choice of method and control variables.

The article uses the global meta-analysis mean of CSA for boys and girls. What are the CIs for these estimates? When evaluating whether the observed risk of CSA is higher (or lower) than the expected, the CI for the expected risk should also be taken under consideration. The CI of the sample estimated mean must be outside of the CI for the global meta-analysis mean.

Why do you use the E/O index instead of a Chi-squared test or Fisher's exact test?

Reviewer 3 Report

This is a critically important paper filling a serious research gap: the prevalence, correlated, and potential sequelae of childhood sexual abuse among indigenous people living in what is currently known as Canada. Descriptive results like these have real merit. However, I have some serious questions about the methods, in particular the representativeness of the sample (which the authors have not quantitatively assessed) and which measures they calculate.

1. First, the manuscript needs a "Table 1" showing the distribution of all of their relevant variables; at a minimum, showing the means and SDs for all continuous variables and prevalence of all levels of all binary or categorical variables, including CSA, its correlates, and its outcomes. This information is currently scattered around the manuscript and is not always complete!

2. Though the authors assert that there is no reason to think their sample is not representative, the demographics of their sample show otherwise: for example, only a third of their sample are cis-heterosexual men, and gender is strongly correlated with CSA risk. Further, everyone sampled had to have had internet and known how to use it (excluding many older people, e.g.) and selects for people who are willing to fill out surveys for small monetary incentives (disproportionately low-income people or people who otherwise have difficulty acquiring work, which also correlates with CSA victimization). This is in addition to the obvious selection bias the authors point out, in which a survey about criminal justice involvement would have likely disproportionately selected people who had experienced CSA given that CSA and criminal legal system contact are correlated. At minimum, the authors must compare the distribution of demographics in their sample to the distribution of demographics among indigenous Canadians writ large. (One easy way to do this would be as a separate column in the "Table 1" I recommended above.)

3. In addition, the authors should include an appendix table showing differences in the demographics and outcomes (as appropriate) between the people who were deleted from this analysis due to, for example, not providing information about their past criminal history, vs. people who were not included. Even if Prime Panels were representative, the heavily redacted sample that the authors actually analyzed may be quite different given the large proportion who were dropped from the analysis.

4. If there are major differences between the sample being analyzed and the national indigenous population, the authors could apply weights to make their sample at least demographically similar to the national population of indigenous Canadians.

5. Global meta-analytic estimates of CSA prevalence are helpful for comparing indigenous Canadians to the world, but those comparisons have drawbacks; in particular, if non-indigenous Canadians have lower rates than the global average, this paper would underestimate inequities between indigenous and non-indigenous Canadians. It might be helpful to compare rates among indigenous Canadians to both the world average and to non-indigenous Canadians.

6. More detail about Prime Panels would be helpful. How does Prime Panels advertise and recruit users?

7. For their current Table 1 (showing E/O and O/E ratios), showing confidence intervals for O/E ratios would help readers understand estimate uncertainty (instead of only showing E/O CIs).

8. For Table 3, given the strong correlation between CSA and childhood poverty, it seems like the authors should present both unconditional odds ratios (which help give a sense of how often indigenous survivors of CSA also experience the selected adult sequelae) and odds ratios conditional on poverty (since, for example, it is hard to say whether the CSA is the driver of these problems or whether poverty is the driver of these problems).

9. In general, much of the conclusions section focuses on speculation that these results would be similar for indigenous people in other former British colonies, with very limited discussion about the potential resources indigenous communities might need to help care for indigenous survivors of CSA or prevent CSA. A greater focus on what these findings suggest about policy or community needs, or on the long-term effects of colonization and why they are rarely studied, is warranted.

Round 2

Reviewer 3 Report

The authors responded to all of my concerns very thoughtfully; the limitations of the survey methods (my biggest concern), and what they may mean for generalizability, are now well described and contextualized. Their new 'implications for policy' section also helps guide readers towards concrete remedies specific to indigenous peoples in Canada.